# Vaccine uptake, associated factors and reasons for vaccination status among the South African elderly; findings and next steps

Mncengeli Sibanda[1,2]*, Rosemary J. Burnett[1,2], Brian Godman[1,3], Johanna C. Meyer[1,2]

**1** Department of Public Health Pharmacy and Management, School of Pharmacy, Sefako Makgatho Health Sciences University, Ga-Rankuwa, Pretoria, South Africa, **2** South African Vaccination and Immunisation Centre, Sefako Makgatho Health Sciences University, Ga-Rankuwa, Pretoria, South Africa, **3** Department of Pharmacoepidemiology, Strathclyde Institute of Pharmacy and Biomedical Sciences, University of Strathclyde, Glasgow, United Kingdom

* mncengeli.sibanda@smu.ac.za

**Data Availability Statement:** All relevant data are within the manuscript and its Supporting Information files.

## Abstract

### Objectives

The elderly are particularly prone to complications from a number of vaccine-preventable diseases. However, there are limited data on vaccine uptake for this vulnerable population in South Africa. Consequently, this study investigated influenza, pneumococcal and shingles vaccine uptake among elderly people in South Africa; reasons for their vaccination status; and factors associated with their uptake.

### Methods

Cross-sectional study using an interviewer-administered questionnaire to survey 985 consenting adults aged ≥65 years in 2018. Participants were recruited from across South Africa. Bivariate analysis was used to identify socio-demographic variables associated with vaccine uptake, with multivariate logistic regression analysis used to identify key factors associated with vaccine uptake.

### Results

Influenza vaccine uptake was 32.3% (318/985), with uptake highest in those aged 85–90 years. Pneumococcal and shingles vaccine uptake was 3.8% (37/985) and 0.4% (4/985) respectively, being highest among those aged >90 years. The strongest statistically significant predictors for influenza vaccination were previous influenza vaccination (OR: 8.42 [5.61–12.64]); identifying as 'Coloured' (OR: 8.39 [3.98–17.69]); and residing in Gauteng Province (OR: 5.44 [3.30–9.02]). The strongest statistically significant predictors of receiving pneumococcal vaccination included receiving influenza vaccination (OR = 10.67 [3.27–37.83]); residing in the Western Cape Province (OR: 7.34 [1.49–36.22]); identifying as 'Indian' (OR: 5.85 [2.53–13.55]); and having a university education (OR: 5.56 [1.25–24.77]). Statistically significant barriers to receiving influenza vaccination included following the Traditional African religion (OR: 0.08 [0.01–0.62]) and residing in Limpopo Province (OR: 0.16

**Funding:** The work reported herein was made possible through funding by the South African Medical Research Council through its Division of Research Capacity Development under the Bongani Mayosi National Health Scholars Programme from funding received from the Public Health Enhancement Fund / South African National Department of Health. The content hereof is the sole responsibility of the authors and does not necessarily represent the official views of the SAMRC. In addition, this study was also funded by the Sefako Makgatho Health Sciences University and the South African National Research Foundation.

**Competing interests:** The South African Vaccination and Immunisation Centre receives unrestricted educational grants from the vaccine industry.

[0.04–0.71]). The main reasons for non-vaccination were considering influenza as a mild illness (36.6%; 242/661), and lack of knowledge about the pneumococcal (93.4%; 886/948) and shingles (95.2%; 934/981) vaccines.

## Conclusion

Vaccine uptake for all vaccines was sub-optimal, with multiple non-modifiable factors predicting vaccine uptake. These pre-COVID-19 data provide a baseline for measuring the effectiveness of future interventions to increase vaccine uptake and safeguard the health of the elderly.

## Introduction

The immune system in adults deteriorates with age following a steady functional decline known as immunosenescence [1, 2]. As a result, humoral and cellular immunological responses are reduced, making it more challenging for the elderly ($\geq$65 years) to combat infections [1–3]. Ageing is also accompanied with a rise in multimorbidity including chronic conditions such as diabetes, malignancies, dementia, renal and cardiovascular diseases affecting physical and mental health [4–6]. Multimorbidity in old age is also a risk factor for the emergence of infectious diseases [7–9]. For example, in the elderly, there is a decrease in gastric acid production and a deterioration in lung mucociliary clearance, both of which may make this population more susceptible to acquiring pneumonia [9]. The elderly are thus generally more susceptible to infectious diseases, including influenza, community acquired pneumonia, coronavirus disease 2019 (COVID-19) and shingles, resulting in hospitalisation, chronic disease and death [7, 10, 11]. Consequently, infectious diseases need be avoided where possible in this vulnerable population through strategies including life-course vaccination [12].

A life-course vaccination strategy will increase overall well-being by making the population healthier and enhancing the quality of life for all age groups including the elderly [2, 12, 13]. Vaccination has been identified as one of the critical strategies for healthy ageing and elevating the quality of life among the elderly, as well as positively affecting national economies and health systems, particularly in the era of population ageing [2, 13, 14]. Age-related immunosenescence may also potentially reduce the effectiveness of vaccines. However, vaccines are still very beneficial in preventing infectious diseases among the elderly, thereby reducing morbidity and mortality; consequently, vaccination should be encouraged [1, 2, 15].

The World Health Organization (WHO) and health authorities in various countries have made recommendations on vaccines for the elderly; however, not even all high-income countries (HICs) have comprehensive adult vaccination schedules [13, 16]. As a result, vaccination programmes currently differ appreciably in HICs in terms of the type and number of vaccines, the targeted population and whether or not the vaccines are available free of charge even to the uninsured in countries without universal healthcare [13, 16]. This situation is much worse in low- and middle-income countries (LMICs) [14, 17]. Where comprehensive adult vaccination schedules exist, influenza, pneumococcal and shingles vaccines, and most recently the COVID-19 vaccines, are commonly recommended for the elderly [12, 13, 16, 18]. In addition, other vaccines used in the general population including tetanus, diphtheria and pertussis may also be recommended in this population [13, 15, 19]. Life-course vaccination is a vital pillar of antimicrobial stewardship, reducing the incidence of infectious diseases in this population and subsequent over-prescribing of antibiotics, thereby decreasing antimicrobial resistance (AMR) [20, 21]. Appreciable over-use of antibiotics was seen with the recent COVID-19 pandemic

especially among LMICs including African countries [22–24]. The greatest burden of AMR is currently seen in sub-Saharan Africa, which urgently needs to be addressed to reduce future morbidity, mortality and costs [25, 26].

Where data are available, the WHO goal of 75% influenza vaccine uptake for the elderly was not reached by most countries by the year 2014/2015 [27]. Improving vaccine uptake by the elderly can decrease the burden of vaccine-preventable diseases (VPDs), thereby reducing morbidity, hospital admissions, health costs, and mortality [2, 12, 19, 28–30], in addition to reducing the over-prescribing of antibiotics and the implications for AMR [21]. Alongside this, high vaccine uptake in the population may further reduce morbidity and mortality from VPDs through herd immunity [28, 31]. Aside from saving lives and reducing the disease burden, there is growing evidence of the effectiveness and cost-effectiveness of influenza, pneumococcal [32–38] and shingles [39] vaccination of the elderly.

In South Africa, currently only a limited number of vaccines have been approved for use in the elderly [18]. These include inactivated trivalent and quadrivalent influenza vaccines; pneumococcal conjugate vaccine (PCV13); the pneumococcal polysaccharide vaccine (PPSV23); COVID-19 vaccines; the live attenuated shingles vaccine; and vaccines combining tetanus toxoid, reduced diphtheria toxoid, and acellular pertussis (TdaP) [18, 40]. However, currently the South African Standard Treatment Guidelines and Essential Medicines List (STGs/EML) only recommends free influenza and COVID-19 vaccination for the elderly [41, 42]. Also, for special conditions such as asplenia or a CSF leak, a free booster dose of PPSV23 is recommended for the elderly who received PPSV23 before the age of 65 years [42]. In contrast, the South African private sector guidelines for vaccination of the elderly are more extensive, making recommendations for pneumococcal vaccination with both PCV13 and PPSV23, in addition to influenza and COVID-19 vaccinations [43, 44]. Unfortunately, the majority of elderly South Africans do not have medical aid or medical insurance cover; consequently, many cannot afford to purchase pneumococcal vaccines from the private sector [18].

Despite efforts to reduce the burden of VPDs through vaccination and other interventions, the incidence of these diseases, especially respiratory tract infections, is high among the South African elderly [45]. For example, between 2013 and 2015, an estimated 36.1% (4 195/11 621) of influenza-associated deaths were in individuals aged ≥65 years [46]. To the best of our knowledge, apart from COVID-19 vaccines, there has been no published data on vaccine uptake among the elderly in South Africa using the public healthcare system. Also, data on privately insured elderly South Africans is confined to one study on influenza vaccine uptake [47]. It is also crucial to understand the reasons for, and factors associated with vaccine uptake among the elderly, as these data will inform the design of interventions to increase vaccine uptake and reduce subsequent morbidity and mortality in this vulnerable population where concerns have been identified [48, 49]. This is especially important given the current extent of vaccine hesitancy and rising AMR rates across Africa [31, 50, 51]. Consequently, this study investigated the uptake of influenza, pneumococcal and shingles vaccines, reasons for being vaccinated or unvaccinated, and socio-demographic factors associated with vaccination status, among the elderly in South Africa. The findings can be used to help develop future strategies in South Africa and beyond to improve the health of the elderly population, with few studies from Africa currently published in this area.

## Methods

### Study design and sampling

A cross-sectional study was conducted using an interviewer-administered, face-to-face, structured questionnaire. The study was conducted in 18 public sector community health centres

(CHCs) and 44 old age homes (OAHs] managed by the private sector and non-governmental organisations, across all nine South African provinces between August 2018 and December 2018. This study was part of a larger study titled 'Vaccination uptake amongst geriatrics in South Africa: A multi-centre study'. Sampling of the study sites has been fully described in a previous publication [52]. CHCs were chosen rather than primary healthcare centres as they provide a comprehensive range of ambulatory care services to the elderly in South Africa for the public healthcare system including vaccinations [52], with OAHs providing continuous long-term assisted living or frail care services to the elderly with physical or mental frailty [52].

All eligible individuals consulting at the CHCs or residing at the selected OAHs were invited to participate in the study. Eligible individuals were defined as consenting adults, aged ≥65 years on the day of data collection, who were able to comprehend the questions. Comprehension was determined by self-proclaimed fluency in at least one of South Africa's official languages and the successful completion of the abbreviated mental test (AMT4), a brief instrument used to assess cognitive ability [53]. Impaired cognition was indicated by an AMT4 score of less than four [53]. Elderly persons who were in frail care, intensive or critical care or presenting with a serious medical condition (e.g. cancer or other terminal disease) were excluded from the study.

**Data collection instrument and process.** The questionnaire was developed based on published literature combined with consultation with experts in the field of vaccinology [54–56]. Sections of the questionnaire made provision for collection of participants' socio-demographic data; medical data including the presence of chronic condition/s, self-reported uptake of influenza, pneumococcal, and shingles vaccines; and reasons for their current vaccination status. Vaccination status and reasons for vaccination status were assessed using the following questions: *"Prior to this year, have you ever received the flu vaccine?"; "Did you receive the flu vaccine this year?"; "Did you receive at least one type of pneumococcal vaccine?"; "Did you receive at least one dose of shingles vaccine?"* and *"What is the main reason for receiving / not receiving the vaccine?".* The questionnaire was translated from English into vernacular languages and back translated to ensure validity.

The questionnaire was pre-tested among 10 elderly patients at the out-patient department of an academic hospital in Pretoria, South Africa, for comprehension of instructions, clarity/understanding of questions, and time and ease of completion. Adjustments were made based on the results of the pre-testing, after which data were collected by the principal investigator (MS) and a trained research assistant during weekdays (08h00 to 16h00) between August and December 2018.

The study's objectives and ethical considerations (informed consent, confidentiality and privacy) were briefly explained to all potential participants present at the facilities on the day of data collection, following which they were invited to participate in the study.

## Data management and statistical analysis

The collected data were captured by MS and the research assistant using Microsoft Excel® (Microsoft Office, 2016). Data were exported from Microsoft Excel® and cleaned prior to analysis using IBM SPSS Statistics (Version 26). Descriptive statistics were used to summarise data including frequency distributions of categorical variables (socio-demographic data; influenza, pneumococcal and shingles vaccination status; and reasons for vaccination status) and measures of dispersion of age (range, mean and standard deviation [SD]).

Participants with unknown vaccination status were excluded from the inferential statistical analyses which was conducted for each vaccine separately. Pearson's chi-square test was used in bivariate analyses comparing the vaccinated and unvaccinated for the independent variables

sex, age group, province, type of facility (CHC or OAH), religion, distance to the nearest health facility and presence of a chronic condition. In addition, influenza vaccination prior to 2018 was included as an independent variable in the analyses for all vaccines, including influenza vaccine uptake in 2018. Variables with chi-square p-values ≤0.05 (2-tailed) were considered statistically significant and included in a logistic regression model for multivariate analysis. Odds ratios (ORs) with 95% confidence intervals (CI 95%) and chi-square p-values were calculated to identify predictors (factors positively associated, i.e. OR≥1) and barriers (factors negatively associated, i.e. OR<1) and strength of the associations relative to other factors found to be associated with vaccine uptake in the bivariate analyses.

### Ethical considerations

The study protocol was approved by the Sefako Makgatho University Research Ethics Committee (SMUREC/P/36/2018:PG) prior to commencement of the study. Permission to conduct the study at the CHCs was obtained from the Provincial Departments of Health and the facility managers, while permission to conduct the study at OAHs was obtained from the Provincial Departments of Social Development and OAH managers.

Signed informed consent was obtained from participants prior to starting the interview. No data on personal identifiers were collected; facility and participant data were coded. Furthermore, data was stored securely, to protect collected data from unauthorised access and to ensure confidentiality. Data will be stored for a minimum of five years after the results have been published, after which it may be safely destroyed in compliance with university policies.

## Results

### Socio-demographic characteristics and vaccine uptake

A total of 985 participants were interviewed, with a mean age of 74.4 years (SD: 6.7; range: 65.0–95.8) and 63.5% (625/985) being female. More participants (55.5% [547/985]) were recruited from CHCs compared to OAHs (44.5%; 438/985). The majority of participants (82.5% [813/985]) had at least one chronic condition, with hypertension being the most common chronic condition affecting 49.1% (399/813) of participants with chronic conditions, followed by diabetes (19.7% [160/813]), HIV (8.5% [69/813]) and rheumatoid arthritis (8.4% [68/813]). Most (98% [965/985]) participants were recipients of a government-sponsored social grant indicating their low socioeconomic status, and 65.5% (645/985) owned a mobile phone.

Although 64.8% (638/985) of participants received at least one dose of influenza vaccine prior to 2018, influenza vaccine uptake for 2018 was only 32.3% (318/985). The mean age of participants vaccinated against influenza was 75.7 years (SD: 7.4), and 76.1 (SD: 6.8) and 86.2 (SD: 5.4) years for pneumococcal and shingles vaccines respectively. Uptake of at least one type of pneumococcal vaccine and shingles vaccine was 3.8% (37/985) and 0.4% (4/985) respectively. Of those who received the influenza vaccine, 10.1% (32/318) also received the pneumococcal vaccine while only 0.3% (3/985) of participants received all three vaccines. Unknown vaccination status (responded *"do not know/not sure"* to vaccination questions) was reported by 0.61% (6/985) of participants for the influenza vaccine, 3.7% (36/985) for the pneumococcal vaccine and 1.11% (11/985) for the shingles vaccine. Among participants with diabetes, vaccine uptake was 27.5% (44/160), 6.3% (10/160) and 1.3% (2/160) for influenza, pneumococcal and shingles vaccines respectively. Influenza uptake among HIV positive participants was 49.3% (34/69), while 8.7% (6/69) received the pneumococcal vaccine and none received the shingles vaccine. See Table 1 for further details.

**Table 1. Socio-demographic characteristics of participants and vaccine uptake (n = 985\*).**

| Variable | | Total, n (%) | Influenza vaccinated in 2018, n (%) | P value | Pneumococcal vaccinated, n (%) | P value | Shingles vaccinated, n (%) | Wald P value |
|---|---|---|---|---|---|---|---|---|
| Total | | 985 | 318 (32.3) | - | 37 (3.8) | - | 4 (0.4) | - |
| Sex | Male | 360 (36.5) | 116 (32.2) | 0.058 | 6 (1.7) | 0.015 | 3 (0.8) | 0.228 |
| | Female | 625 (63.5) | 202 (32.3) | | 31 (5.0) | | 1 (0.2) | |
| Facility type | Community health centre | 547 (55.5) | 176 (32.2) | 0.389 | 17 (3.1) | 0.489 | 1 (0.2) | 0.243 |
| | Old age home | 438 (44.5) | 142 (32.4) | | 20 (4.6) | | 3 (0.7) | |
| Age (years) | 65–70 | 322 (32.7) | 85 (26.4) | <0.001 | 7 (2.2) | 0.013 | 0 (0.0) | <0.001 |
| | 70–75 | 248 (25.2) | 72 (29.0) | | 10 (4.0) | | 0 (0.0) | |
| | 75–80 | 209 (21.2) | 69 (33.0) | | 13 (6.2) | | 0 (0.0) | |
| | 80–85 | 135 (13.7) | 54 (40.0) | | 3 (2.2) | | 2 (1.5) | |
| | 85–90 | 49 (5.0) | 28 (57.1) | | 1 (2.0) | | 0 (0.0) | |
| | 90+ | 22 (2.2) | 10 (45.5) | | 3 (13.6) | | 2 (9.1) | |
| Marital status | Married | 275 (27.9) | 78 (28.4) | <0.001 | 9 (3.3) | <0.001 | 0 (0.0) | 0.646 |
| | Widowed | 485 (49.2) | 159 (32.8) | | 23 (4.7) | | 4 (0.8) | |
| | Divorced | 65 (6.6) | 38 (58.5) | | 0 (0.0) | | 0 (0.0) | |
| | Living with partner | 15 (1.5) | 3 (20.0) | | 0 (0.0) | | 0 (0.0) | |
| | Single | 145 (14.7) | 40 (27.6) | | 5 (3.4) | | 0 (0.0) | |
| Race | Black | 505 (51.3) | 104 (20.6) | <0.001 | 7 (1.4) | <0.001 | 0 (0.0) | 0.310 |
| | Coloured | 35 (3.6) | 24 (68.6) | | 0 (0.0) | | 0 (0.0) | |
| | White | 380 (38.6) | 180 (47.4) | | 28 (7.4) | | 4 (1.1) | |
| | Indian | 65 (6.6) | 10 (15.4) | | 2 (3.1) | | 0 (0.0) | |
| Province | Eastern Cape | 92 (9.3) | 14 (15.2) | <0.001 | 3 (3.3) | 0.004 | 0 (0.0) | <0.001 |
| | Free State | 100 (10.2) | 13 (13.0) | | 6 (6.0) | | 0 (0.0) | |
| | Gauteng Province | 150 (15.2) | 87 (58.0) | | 6 (4.0) | | 0 (0.0) | |
| | KwaZulu-Natal | 158 (16.0) | 32 (20.2) | | 2 (1.3) | | 0 (0.0) | |
| | Limpopo Province | 50 (5.1) | 2 (4.0) | | 0 (0.0) | | 0 (0.0) | |
| | Mpumalanga Province | 135 (13.7) | 47 (34.8) | | 8 (5.9) | | 0 (0.0) | |
| | Northern Cape | 80 (8.1) | 14 (17.5) | | 0 (0.0) | | 0 (0.0) | |
| | North West | 140 (14.2) | 69 (49.3) | | 5 (3.6) | | 0 (0.0) | |
| | Western Cape | 80 (8.1) | 40 (50.0) | | 7 (8.8) | | 4 (5.0) | |

(*Continued*)

**Table 1.** (Continued)

| Variable | | Total, n (%) | Influenza vaccinated in 2018, n (%) | P value | Pneumococcal vaccinated, n (%) | P value | Shingles vaccinated, n (%) | Wald P value |
|---|---|---|---|---|---|---|---|---|
| Education level | No education | 232 (23.6) | 53 (22.8) | <0.001 | 5 (2.2) | 0.012 | 0 (0.0) | 0.532 |
| | Primary (not completed) | 245 (24.9) | 59 (24.1) | | 4 (1.6) | | 1 (0.4) | |
| | Primary (completed) | 179 (18.2) | 35 (19.6) | | 9 (5.0) | | 1 (0.5) | |
| | Secondary (completed) | 302 (30.7) | 159 (52.6) | | 16 (5.3) | | 2 (0.6) | |
| | University | 27 (2.7) | 12 (44.4) | | 3 (11.1) | | 0 (0.0) | |
| Chronic condition | Yes | 813 (82.5) | 256 (31.5) | 0.003 | 25 (3.1) | 0.011 | 2 (0.2) | 0.156 |
| | No | 172 (17.5) | 62 (36.0) | | 12 (7.0) | | 2 (1.2) | |
| Religion | Christian | 932 (94.6) | 308 (33.0) | 0.042 | 37 (4.0) | 0.344 | 4 (0.4) | 0.929 |
| | Muslim | 28 (2.8) | 9 (32.1) | | 0 (0.0) | | 0 (0.0) | |
| | Traditional African | 25 (2.5) | 1 (4.0) | | 0 (0.0) | | 0 (0.0) | |
| Distance to nearest health facility | 0–5 km | 763 (77.5) | 233 (30.5) | 0.004 | 24 (31.4) | 0.363 | 3 (0.4) | 0.458 |
| | 5–10 km | 122 (12.4) | 44 (36.1) | | 6 (4.9) | | 1 (0.8) | |
| | 10–15 km | 100 (10.2) | 41 (41.0) | | 7 (7.0) | | (0.0) | |

*The sample size of 985 was used as a denominator for descriptive statistical analysis and not for inferential statistics

## Reasons for vaccine decisions

The most common reason given by participants for receiving the influenza vaccine was to protect themselves against illness (55.7% [177/318]). Lack of access to the vaccine (i.e. vaccine not available at facility) was the main reason for not receiving the influenza vaccine (23.4% [155/661]), followed by reasons related to vaccine hesitancy (31.9% [211/661]). For both the pneumococcal vaccine and the shingles vaccine, the main reason for not having received the vaccine was a lack of knowledge about the vaccine/s, 96.6% (881/912) and 96.2% (933/970) respectively. Of those who were aware of the pneumococcal and shingles vaccines and were not vaccinated, cost was a major obstacle for 23.8% (8/31) and 37.8% (14/37) respectively. Table 2 shows further details of the reasons for receiving or not receiving the vaccines respectively among the participants.

## Factors associated with influenza and pneumococcal vaccination

Socio-demographic variables that were identified by the Pearson's chi-square test to be statistically significantly associated with vaccination were further analysed using logistic regression analysis. Previous influenza vaccination (prior to 2018) (OR:8.42) was the strongest predictor of influenza vaccine uptake in 2018. This was followed by identifying as 'Coloured' (OR: 8.39); and residing in Gauteng Province (OR: 5.44). Statistically significant barriers to receiving influenza vaccination included following the Traditional African religion (OR: 0.08) and residing in Limpopo Province (OR: 0.16). See Table 3 for further details.

Influenza vaccine uptake in 2018 was also a strong predictor of pneumococcal vaccine uptake (OR: 10.67). Residing in the Western Cape Province (OR: 7.34); identifying as 'Indian'

**Table 2. Frequency distribution of main reasons for vaccine decisions.**

| Reasons for vaccination | Type of vaccination; n (%) | | |
|---|---|---|---|
| | Influenza (n = 318) | Pneumococcal (n = 37) | Shingles (n = 4) |
| To protect themselves against illness | 177 (55.7) | 8 (21.6) | - |
| Recommendation by their healthcare worker | 70 (22.0) | 17 (45.9) | 1 (25.0) |
| Pre-existing condition | 65 (20.4) | 12 (32.4) | 3 (75.0) |
| To obtain Vitality Points[b] | 6 (1.9) | - | - |
| **Reasons for non-vaccination** | **Influenza (n = 661)** | **Pneumococcal (n = 912)** | **Shingles (n = 970)** |
| **Lack of information** | | | |
| Lack of knowledge about vaccine | - | 881 (96.6) | 933 (96.2) |
| **Reasons related to vaccine hesitancy** | | | |
| Fear of side effects/reactions | 136 (20.6) | - | - |
| Do not trust that vaccine is effective | 47 (7.1) | - | - |
| Believe illness is mild / being naturally immune to disease | 7 (1.1) | - | 2 (0.2) |
| Religious reasons | 21 (3.2) | 13 (1.4) | 12 (1.2) |
| **Lack of motivation** | | | |
| Forgot | 46 (7.0) | - | - |
| **Obstacles** | | | |
| Vaccine not available at facility | 155 (23.4) | - | - |
| Vaccine too expensive | - | 8 (0.9) | 14 (1.4) |
| Allergic to vaccine or vaccine contraindicated | 14 (2.1) | 10 (1.1) | 9 (0.9) |

[b]Vitality Points are rewards from a medical insurance loyalty programme awarded to the beneficiary after they complete educational, fitness, healthy living and/or preventative activities toward the achievement of wellness goals.

(OR: 5.85); and having a university education (OR: 5.56) were also significant predictors of pneumococcal vaccine uptake.

Multivariate analysis was not undertaken to identify factors associated with shingles vaccination as the number of participants who received the shingles vaccine was less than the minimum required for such analysis [57].

## Discussion

To the best of our knowledge, this is the first study that investigated the uptake of three vaccines recommended for the elderly (influenza, pneumococcal and shingles vaccines) in South Africa, the reasons for their vaccination status, and factors associated with vaccine uptake. Since the study was conducted prior to the COVID-19 pandemic, the findings provide a baseline against which post-pandemic changes in vaccine uptake can be measured, to identify both increases and reductions among the elderly as reported in other parts of the world [58–60]. This is important since there are still a number of infectious diseases affecting the elderly where effective vaccines, such as respiratory syncytial virus vaccines which are available in HICs [61], have not yet been licensed in South Africa, or are still under development [15]. Consequently, it is important to continually ascertain current vaccine uptake in the vulnerable elderly population, and the reasons for sub-optimal uptake, to instigate additional initiatives if needed [15].

Based on our findings, about a third of participants had been vaccinated against influenza in 2018 whilst a very small proportion were vaccinated against pneumococcal disease and shingles. This finding was expected since only the influenza vaccine is routinely recommended for all adults aged ≥65 years in the STGs/EML and freely available in the public sector, with the

**Table 3. Multivariate model of factors related to influenza and pneumococcal vaccine uptake.**

| Variable | | Influenza vaccination in 2018 | | Pneumococcal vaccination | |
|---|---|---|---|---|---|
| | | OR (95% CI) | P value | OR (95% CI) | P value |
| **Age (years)** | **65–70** | Reference | | Reference | |
| | **70–75** | 1.14 (0.79–1.64) | 0.499 | 1.88 (0.71–5.03) | 0.206 |
| | **75–80** | 1.41 (0.96–2.06) | 0.078 | 3.01 (1.18–7.67) | 0.021 |
| | **80–85** | 1.87 (1.23–2.87) | 0.004 | 1.08 (0.28–4.26) | 0.908 |
| | **85–90** | 3.70 (2.00–6.87) | 0.001 | 0.95 (0.11–7.90) | 0.962 |
| | **90+** | 2.31 (0.96–5.55) | 0.06 | 7.29 (1.74–30.56) | 0.07 |
| **Marital status** | **Married** | 0.81 (0.58–1.12) | 0.194 | 0.68 (0.31–1.48) | 0.328 |
| | **Widowed** | Reference | | Reference | |
| | **Divorced** | 3.37 (1.94–5.84) | <0.001 | 0 (0.0) | 0.997 |
| | **Living with partner** | 0.51 (1.42–1.83) | 0.301 | 0 (0.0) | <0.001 |
| | **Single** | 0.78 (0.52–1.17) | 0.227 | 0.73 (0.27–1.97) | 0.539 |
| **Race** | **Black** | Reference | | Reference | |
| | **Coloured** | 8.39 (3.98–17.69) | <0.001 | 0 (0.0) | <0.001 |
| | **White** | 0.70 (0.35–1.42) | 0.322 | 2.23 (0.45–10.98) | 0.324 |
| | **Indian** | 3.55 (2.64–4.77) | <0.001 | 5.85 (2.53–13.55) | <0.001 |
| **Province** | **Eastern Cape** | 0.71 (0.36–1.40) | 0.323 | 2.60 (0.43–15.89) | 0.300 |
| | **Free State** | 0.60 (0.30–1.20) | 0.147 | 4.98 (0.98–25.19) | 0.052 |
| | **Gauteng Province** | 5.44 (3.30–9.02) | <0.001 | 3.33 (0.66–16.78) | 0.145 |
| | **KwaZulu-Natal** | Reference | | Reference | |
| | **Limpopo Province** | 0.16 (0.04–0.71) | 0.016 | 0 (0.0) | <0.001 |
| | **Mpumalanga Province** | 2.20 (1.30–3.73) | 0.03 | 5.25 (1.09–25.20) | 0.038 |
| | **Northern Cape** | 0.84 (0.42–1.67) | 0.612 | 0 (0.0) | 0.998 |
| | **North West** | 3.83 (2.30–6.37) | <0.001 | 2.84 (0.54–14.87) | 0.217 |
| | **Western Cape** | 4.04 (2.24–7.27) | <0.001 | 7.34 (1.49–36.22) | 0.014 |
| **Education level** | **No education** | Reference | | Reference | |
| | **Primary (not completed)** | 1.05 (0.69–1.61) | 0.811 | 0.73 (0.19–2.74) | 0.635 |
| | **Primary (completed)** | 0.81 (0.50–1.31) | 0.073 | 2.38 (0.78–7.24) | 0.126 |
| | **Secondary** | 3.67 (2.51–5.38) | <0.001 | 2.43 (0.88–6.75) | 0.087 |
| | **University** | 2.64 (1.17–5.99) | 0.020 | 5.56 (1.25–24.77) | 0.025 |
| **Chronic condition** | **Yes** | Reference | | Reference | |
| | **No** | 1.27 (0.90–1.79) | 0.179 | 2.43 (1.20–4.96) | 0.014 |
| **Religion** | **Christian** | Reference | | Reference | |
| | **Muslim** | 0.95 (0.43–2.13) | 0.901 | 0 (0.0) | <0.001 |
| | **Traditional African** | 0.08 (0.01–0.62) | 0.015 | 0 (0.0) | <0.001 |
| **Distance to nearest health facility** | **0–5 km** | Reference | | Reference | |
| | **5–10 km** | 1.78 (1.16–2.72) | 0.008 | 1.93 (0.77–4.84) | 0.162 |
| | **10–15 km** | 1.19 (0.79–1.79) | 0.406 | 1.88 (0.79–4.47) | 0.152 |
| **Previous influenza vaccination** | **Yes** | 8.42 (5.61–12.64) | <0.001 | 10.67 (3.27–34.83) | <0.001 |
| | **No** | Reference | | Reference | |

pneumococcal vaccine only recommended in special circumstance [18]. The proportion of participants who received all three vaccines was exceedingly low, with no significant difference in the uptake between those attending CHCs and those living in OAHs.

In South Africa, several socio-economic inequalities in health and healthcare exist [62]. Gauteng and Western Cape are amongst the wealthiest provinces (first and third highest contributor to the total gross domestic product) and most racially diverse provinces in South

Africa [63]. Furthermore, these two provinces have made significant investments in healthcare and health infrastructure [64, 65]. Race and geographic location have a significant influence on individual income, level of education, demand for healthcare, and the supply and standard of healthcare [66–68]. These health inequalities, which are strongly linked to the social determinants of health, could possibly explain the differences in vaccine uptake across geographic location (provinces), race and other socio-demographic variables.

## Influenza vaccination

In general, there are more published studies on influenza vaccine uptake among the elderly than for any other VPD affecting the elderly, with vaccine uptake differing widely across countries. Published uptake by the elderly ranges from as low as 18.5% up to as high as 83.3%, with higher uptake principally seen among HICs [69–85]. Our finding of 32.3% uptake is in line with some of the findings from HICs in Europe including Austria, Slovenia and Germany, and a number of Central and Eastern European countries in 2022 [83]. Encouragingly, our study found much higher influenza vaccine uptake than a previous South African study reporting only 18.5% uptake among elderly private health insurance scheme members in 2015 [47]. However, 32.3% is significantly lower than the WHO's recommended target of 75% [27].

Overall, as mentioned, there are few published studies assessing influenza vaccine uptake among the elderly in Africa and other LMICs. Other than Solanki et al, 2018 [47], no publications could be found from the WHO African Region, while the few studies conducted in North Africa have reported low influenza uptake by the elderly. For example, a Tunisian study on elderly patients with chronic conditions attending primary and secondary healthcare facilities reported that 19.4% received influenza vaccination during the 2018–2019 influenza season [76]. Similarly only 21.8% of elderly patients admitted to state hospital clinics in Türkiye in 2019 had received influenza vaccination [77]. Influenza vaccine uptake has also been low in China ranging between 0.3% and 0.6% of patients aged >60 years with chronic conditions residing in Shanghai, China [75]. In contrast, high uptake of up to 79.7% has been reported among the elderly in Brazil, although not reaching the 80% target set by the Brazilian Government [72]. The decentralisation and expansion of health services are regarded as essential for the high influenza vaccine uptake and success of the National Immunization Program in Brazil [72]. However, a later study in the city of Rio Grande, Brazil reported only 27.9% uptake by the elderly population [78].

Our study found the need 'to protect themselves', 'recommendation by a healthcare worker (HCW)' and 'a pre-existing condition' to be the commonest reasons for accepting the influenza vaccine. These findings are supported by previous studies conducted in other countries. For example, a Serbian study reported the main reason for influenza vaccination in a cohort of elderly persons was physician recommendation (37.4%) followed by 'prevention of flu' (33.7%) [84]. Similarly, a doctor's recommendation was the main reason (41.1%) for vaccination among elderly Tunisian study participants [76].

The principal reasons for non-vaccination among our study participants was the unavailability of the influenza vaccine followed by reasons related to vaccine hesitancy. Studies from other countries have also reported vaccine hesitancy in the elderly population, including a Serbian study reporting "They were in good health" (33.5%) and "they did not believe that vaccine protects from flu" (31.5%) [84]. In Tunisia, the two principal reasons for not being vaccinated were concerns about side-effects of the vaccine (71.5%) and the perception of the low effectiveness of the influenza vaccine (33.9%) [76]. Furthermore, while the cost of the influenza vaccine was an important reason for non-vaccination among elderly diabetic patients in some countries where co-payments can be an issue [69], this was not the case in our study, since influenza

vaccination is provided free of charge for South Africans using the public health sector. In Shanghai, one of the reasons for poor vaccine uptake was the inconvenience associated with vaccination with CHCs offering vaccine services for adults in one or two half days per week [75]. In addition, a lack of knowledge of the vaccine, mistrust of vaccines and lack of physician recommendation for the vaccine may have also contributed to the low vaccination uptake [75].

This study confirmed previous reports that socio-demographic factors such as ethnicity (white race), increasing age and higher levels of education were positively associated with influenza and pneumococcal vaccine uptake [72, 85, 86]. In addition, previous influenza vaccination was a key predictor of influenza vaccine uptake in our study, similar to others [30, 85]. Since the effectiveness of the influenza vaccine in the elderly has been found to be 58% against laboratory-confirmed influenza and of 41% against influenza-like illness [87], previously vaccinated participants in our study are likely to have experienced fewer influenza infections than their peers, thereby increasing their confidence in influenza vaccination.

## Pneumococcal vaccination

Currently, the WHO has not set a pneumococcal vaccine uptake target for the elderly. However, our finding of only a 3.8% uptake is clearly too low, as is the highest uptake of 13.6% among those aged >90 years. Similar low uptake in the elderly has been reported in other LMICs. For example, only 4.3% of elderly patients admitted to state hospital clinics in Türkiye in 2019 had received the pneumococcal vaccine [77]. This was marginally higher than the 3% of elderly patients with community acquired pneumonia admitted to nine medical centres in Türkiye between 2009 and 2013 [30]. In mainland China, a 2021 systematic review of pneumococcal vaccine uptake among the elderly revealed a similarly low average coverage of only 5.5% (CI 2.4–11.7%) [88]. However, higher uptake was seen in a study conducted in Shanghai in 2017, where 35.7% of those aged 70–79 years with chronic diseases were vaccinated [75]. Uptake is also low in many HICs. For example, only 9.1% of hospitalised elderly diabetics in Poland interviewed in 2013 had received a pneumococcal vaccine [62], while 18% of the elderly who participated in a 2016 survey in nine other European countries had received a pneumococcal vaccine [89]. Similarly, only 15.2% of the elderly with health insurance in northern Israel received PPSV23 from 2010 to 2015 [90]. In contrast, the pneumococcal vaccine uptake among elderly participants in annual household surveys in the United States (US) was 69.0% for one dose in 2017 and 2018, and 32.3% for two or more doses in 2018 [91]. Relatively high uptake by the elderly has also been seen in a national survey conducted in Greece in 2019, at 49.5% and 23.5% for PCV and PPSV23, respectively [82].

Formally adding free pneumococcal vaccines to the vaccination schedule for the elderly could potentially increase uptake of these vaccines as seen in Japan, where two years after PPV23 was added to the national immunisation programme, uptake among the elderly increased from 20.9% to 40.6% [92]. Similarly, since 2005 Australia offers free pneumococcal vaccination to adults aged ≥65 years, with a systematic review of articles published between 1990 and 2015 reporting vaccine uptake increasing from an average of 35.4% prior to 2005 (ranging from 15.4% to 57.9%), to an average of 56.0% from 2005 onwards (ranging from 50.3% to 72.8%) [73]. In 2022, a South African Working Group developed a comprehensive overview of pneumococcal vaccination recommendations for adults in South Africa, to address the perceived uncertainty that most clinicians had regarding use of these vaccines in adults [93]. Through the appropriate use of pneumococcal vaccines, it is hoped that there would be a significant decrease in the morbidity and mortality caused by pneumococcal infections in adults in South Africa [93]. Since a meta-analysis found vaccine effectiveness for PPSV23 to

range between 28.0% to 54.1% for the elderly aged between 65 and 79 and 75.0% for PCV13 in individuals aged ≥65 [94], while receiving PPSV23 at least a year after PCV13 increased effectiveness to 80.3%, it is important that this vaccination schedule be followed in South Africa [95]. We will continue to monitor the situation.

The main reason for non-vaccination among our study participants was lack of knowledge about pneumococcal vaccines. This is similar to a Polish study on elderly diabetics in 2013, and a 2017 study in Shanghai, China, where lack of knowledge of vaccines was one of the major reasons for non-vaccination [69, 75]. In addition, the majority of unvaccinated elderly (98.5%) had never had the pneumococcal vaccine recommended by their physician [69]. Similarly, the 2016 survey of the elderly in nine European countries reported the principal reason (54%) for not being vaccinated was not being offered vaccination by a physician [89]. Thus, while the participants in our study did not mention being offered the vaccine by their HCW, it is likely that this may be why they lacked knowledge about the pneumococcal vaccine. Furthermore, the low vaccine uptake could have been exacerbated by poor access in the public sector, a lack of guidelines for routine vaccination of the elderly against pneumococcal disease and the high cost of PCV in the private sector for uninsured adults. We will be exploring this further in future studies, given the low vaccine uptake identified by our study coupled with the high burden of pneumonia in the elderly [89].

Influenza vaccine uptake was the greatest predictor of pneumococcal vaccine uptake in this study, similar to other studies [30, 85, 96, 97]. Chronic conditions such as diabetes, chronic respiratory diseases and HIV, make a person more susceptible to influenza, pneumonia, and other infectious illnesses, and previous studies have identified an association between pneumococcal vaccine uptake and the presence of chronic conditions justifying its administration [75, 96, 98]. However, in our study, not having a chronic illness was a predictor of pneumococcal vaccine uptake. While this may point to a lack of knowledge among HCWs as highlighted by the South African Working Group [93], further research is needed to clarify this anomaly and determine why this was the case in our study, coupled with the need for a greater understanding of the reasons behind current low vaccination rates. We will be following this up in the future.

## Shingles vaccination

The very low shingles vaccine uptake among the elderly reported in our study is similar to other studies from LMICs including Türkiye at 4.8% [77]. Low shingles vaccination coverage has also been reported from HICs, for example, only 7.7% of those aged >50 years in Saudi Arabia [99]; 8.4% of the elderly in Alberta, Canada [100]; and 11% of the elderly in Greece in 2023 [101], which was substantially lower than the 20% reported in a similar population for 2019 [92]. In 2020 the sale of the live attenuated shingles vaccine was stopped in the US, where uptake had ranged from 7% to 35% [102], with the national surveillance programme reporting an increase from 24.2% in 2013 to 34.2% in 2017 (the year the recombinant shingles vaccine became available in the US) among adults aged >60 years [91]. The change to using the recombinant vaccine was based on superior efficacy data, with a recent systematic review and meta-analysis reporting 94.0% efficacy against shingles amongst immunocompetent elderly persons, and 91.2% efficacy against postherpetic neuralgia in adults aged ≥50 years [103]. Thus, currently only the recombinant shingles vaccine is available, with only 17% coverage being reported from October 2017 to January 2021, suggesting that interventions may be needed to increase uptake [102]. In Australia, shingles vaccination is recommended from the age of 60 years, and in 2016 became available free of charge from the age of 70 years. A 2019 national survey reported that 32% of adults aged ≥65 years had received shingles vaccination, with the highest uptake (55% being in those aged 75–84 years [104]. A similar relatively high

uptake of 53.4% in adults aged ≥70 years was reported from England, where shingles vaccination is offered free of charge to those aged ≥70 years since 2013.; however, lower rates were associated with most ethnic minorities and lower income levels [80]. These findings may be relevant to South Africa and warrant further investigation in our setting.

Similar to our study, studies undertaken in Italy (2014–2015) and France (2018–2019) have also identified lack of knowledge regarding the availability of the shingles vaccine as a major reason for not receiving shingles vaccination [105, 106], with receiving advice from their physician being associated with uptake in the Italian study [105]. It is well recognised that the shingles vaccine is costly, which also has implications for its uptake [107]. Interestingly, a study conducted in Shanghai, China, reported that 16.6% of those aged 50 to 69 years were willing to be vaccinated against shingles, which increased to 72.3% if the vaccine was covered by medical insurance [108].

## Study limitations

Our study has several limitations. Firstly, this study was only undertaken in OAHs and CHCs and did not include elderly persons in the general community. We also utilised convenience sampling; consequently, there is a risk of selection bias. In addition, it can be assumed that elderly persons living in OAHs and attending CHCs receive more care from HCWs than those living in the community and not attending CHCs, which might affect the generalisability of the results.

In South Africa, there is no centralised 'whole of life' vaccine registry and records of vaccines are typically unavailable. As such, the information on vaccine uptake and the reasons for non-vaccination were self-reported, not validated and subject to recall bias.

Furthermore, this study did not distinguish what type of pneumococcal vaccine the participants received (whether PCV or PPSV23), nor did it record the number of pneumococcal doses received.

This study also did not collect data on modifiable factors associated with vaccine uptake, thus the utility of the logistic regression results in generating evidence-based data on which interventions can be based is limited. This analysis was conducted mainly to ensure comparability with similar studies conducted in different settings, in order to identify socio-demographic factors that may be unique in the South African setting. However, data were collected on reasons for vaccination status, thus the interventions suggested in the recommendations are largely based on these data.

Finally, this study was undertaken pre-COVID-19, and it is well established that the pandemic had a significant impact on the uptake of all vaccines. However, despite these limitations, we believe the findings are robust and applicable today in post-pandemic South Africa, by providing guidance to all key stakeholders with an interest in further improving vaccine literacy and uptake in South Africa.

## Conclusion

Influenza vaccines for the elderly are freely available in the South African public sector and it could be argued that this is the reason for their relatively higher uptake compared to other vaccines in this study. Complacency regarding influenza illness was the main reason for low uptake. In contrast, lack of knowledge about the existence of the pneumococcal and shingles vaccines led to low uptake of these vaccines. The COVID-19 pandemic highlighted the importance of adult vaccination, specifically vaccination for the elderly who were the second most prioritised group after HCWs during the phased roll-out of the COVID-19 vaccination programme in South Africa. As this study was conducted before the COVID-19 pandemic,

vaccine literacy amongst the elderly may have since improved, but on the other hand, vaccine hesitancy may have increased. Thus, vaccine uptake for other important vaccines for the elderly such as influenza, pneumococcal and shingles vaccines, may have changed. This study provides a baseline for measuring this change, and for measuring the effectiveness of future interventions to reduce the burden of infectious diseases in the elderly, thereby reducing inappropriate prescribing of antibiotics, in accordance with the National Action Plan in South Africa to reduce AMR.

## Recommendations

Extending free access to other vaccines recommended for the elderly, in particular the pneumococcal and shingles vaccines, is likely to increase uptake by the elderly. Economic modelling including cost-effectiveness and budget impact analyses need to be conducted for the South African context to determine whether universal free vaccines (pneumococcal, shingles and other vaccines, in addition to influenza vaccine) should be recommended for all adults aged ≥65 years, or if this should be confined to only high-risk adults. This is especially important in the context of population ageing as South Africa progresses towards achieving universal health coverage.

The availability of free vaccines for the elderly should be accompanied by interventions to ensure that there is high public demand and uptake of these vaccines. These include educational efforts for all stakeholders (elderly patients, their guardians, their HCWs and caregivers), focusing on the benefits of vaccines and their role in healthy ageing. The use of health technologies such as electronic reminders integrated into mobile health and decision-making tools built into electronic medical records should be a future priority. For example, the electronic vaccination data system (EVDS) that was used for the South African COVID-19 vaccination programme to issue appointments, reminders and vaccination certificates [109] could be expanded on to include all vaccinations throughout the life course. In addition, these health technologies could potentially serve as a platform for aiding dissemination of vaccine related information for the elderly, as well as for tracing and monitoring of vaccine safety surveillance and other post-vaccination services to help address concerns. A feature of the South African COVID-19 vaccination programme was that vaccine services were brought to the elderly for their convenience and to ensure maximal utilisation of the vaccine [110]. Similarly, bringing vaccination services for all other vaccines to OAHs where the elderly reside and locations where the elderly frequent, such as churches or collection points where the elderly receive their monthly social grants, may also increase vaccine uptake.

## Supporting information

**S1 File.**
(XLSX)

## Author Contributions

**Conceptualization:** Mncengeli Sibanda, Rosemary J. Burnett, Johanna C. Meyer.

**Data curation:** Mncengeli Sibanda, Rosemary J. Burnett, Brian Godman, Johanna C. Meyer.

**Formal analysis:** Mncengeli Sibanda.

**Investigation:** Mncengeli Sibanda.

**Methodology:** Mncengeli Sibanda, Rosemary J. Burnett, Johanna C. Meyer.

**Project administration:** Mncengeli Sibanda, Johanna C. Meyer.

**Supervision:** Rosemary J. Burnett, Brian Godman, Johanna C. Meyer.

**Validation:** Mncengeli Sibanda, Brian Godman, Johanna C. Meyer.

**Writing – original draft:** Mncengeli Sibanda.

**Writing – review & editing:** Rosemary J. Burnett, Brian Godman, Johanna C. Meyer.

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
