## [Decision Letter · Decision Letter 0]

24 Apr 2024

PONE-D-24-06704Vaccine uptake, associated factors and reasons for vaccination status among the South African elderly; findings and next stepsPLOS ONE

Dear Dr. Sibanda,

Thank you for submitting your manuscript to PLOS ONE. After careful consideration, we feel that it has merit but does not fully meet PLOS ONE’s publication criteria as it currently stands. Therefore, we invite you to submit a revised version of the manuscript that addresses the points raised during the review process.Please fully address all comments by the two reviewers and submit the revised manuscript for another round of review. 

We look forward to receiving your revised manuscript.

Kind regards,

Sanjai Kumar

Academic Editor

PLOS ONE

Journal Requirements:

"Sefako Makgatho Health Sciences University, the South African Medical Research Council 583 and the South African National Research Foundation jointly funded this study."

"I have read the journal's policy and the authors of this manuscript have the following competing interests: South African Vaccination and Immunisation Centre receives unrestricted educational grants from the vaccine industry."

5. We note that your Data Availability Statement is currently as follows: [All relevant data are within the manuscript and its Supporting Information file]

Reviewers' comments:

Reviewer's Responses to Questions

**Comments to the Author**

1. Is the manuscript technically sound, and do the data support the conclusions?

Reviewer #1: Partly

Reviewer #2: Yes

2. Has the statistical analysis been performed appropriately and rigorously? 

Reviewer #1: Yes

Reviewer #2: I Don't Know

3. Have the authors made all data underlying the findings in their manuscript fully available?

Reviewer #1: Yes

Reviewer #2: Yes

4. Is the manuscript presented in an intelligible fashion and written in standard English?

Reviewer #1: Yes

Reviewer #2: Yes

5. Review Comments to the Author

Reviewer #1: The authors’ study provides important insight into vaccine uptake among the elderly living in public community health centers and old age homes in South Africa. The lack of a previous baseline for measuring vaccine uptake in the elderly, highlights the importance of this work. The authors find that the influenza, pneumococcal and shingles vaccine uptake was 32.3%, 3.8% and 0.4% respectively. The reasons for vaccination and non-vaccination were different between Influenza vaccination and pneumococcal and shingles vaccination. The main reason given for influenza vaccination was to protect themselves against illness, while the main reason for non-vaccination was vaccine availability and vaccine hesitancy. For pneumococcal and shingles vaccination, the main reasons for non-vaccination were the lack of knowledge about the vaccine. In a multivariate analysis, prior influenza vaccination was a strong predictor of 2018 influenza vaccine uptake while Influenza vaccine uptake in 2018 was a strong predictor of pneumococcal vaccine uptake. Other factors were identified in Table 3, but not expanded upon in the results section. The manuscript is well written, and the authors contextualized their results with similar studies from both HICs and LMICs. However, the authors could have expanded their reporting of their multivariate analysis. Additionally, some of the authors' recommendations appear to not be fully supported by their own data.

Comments

1. The authors performed a multivariate analysis to identify factors associated with uptake of influenza and pneumococcal vaccination. In the abstract, the authors identify significant predictors of vaccine uptake, such as residing in the Western Cape Province, but do not address the same predictors in the results section, except for in Table 3. The authors should describe in more details the significant predictors they identified.

2. The authors should provide more context on why different residing in different provinces, race and education may contribute to increased odds of receiving a vaccine. The factors were highlighted in the abstract and reported in Table 3 but were not contextualized in the discussion section.

3. The authors recommend extending free access to other vaccines, including the pneumococcal vaccine. The authors should define what “other vaccines” they believe should be free. Additionally, their own data does not seem support their recommendation where they report that only 0.9% and 1.4% of individuals who did not receive the pneumococcal and shingles vaccine identified cost as a major obstacle. The authors should support their recommendations with data from their own study and/or provide more context to why the recommendation should be given considering their own results.

4. In the recommendation section the authors recommend the use of health technologies such as electronic medical records. The authors should provide the reason for this recommendation.

5. The authors highlight a feature of the South African COIVID-19 vaccination program where vaccine services were brought to the elderly. The authors should provide citations for the program.

Reviewer #2: Summary

This study investigated the uptake of influenza, pneumococcal and shingles vaccines, reasons for being vaccinated or unvaccinated, and socio-demographic factors associated with vaccination status, among the elderly in South Africa. As the authors state, there are few published studies assessing influenza vaccine uptake among the elderly in Africa and other LMICs indicating the importance of this study.

This study reported a 32.3% uptake of influenza vaccine in the elderly in South Africa. This uptake was higher than a previous South African study reporting an influenza vaccine uptake of 18.5% in 2015 but much lower than the WHO’s recommended target of 75%. The two major reasons for vaccination were “the need to protect themselves” and “a pre-existing condition.” The principal reasons for non-vaccination were unavailability of the influenza vaccine and vaccine hesitancy. The study also confirmed that socio-demographic factors such as ethnicity (white race), increasing age, and higher levels of education were positively associated with vaccine uptake.

The study also reported a 3.8% vaccine uptake of pneumococcal vaccine. The main reason for non-vaccination among study participants was a lack of knowledge about the pneumococcal vaccines. Influenza vaccine uptake was the greatest predictor of pneumococcal vaccine uptake in this study.

Specific Comments

1. Please describe the efficacy of the three types of vaccines (influenza, pneumococcal, and shingles) studied.

2. Please discuss why the rate of influenza vaccination is so low in Shanghai and so high in Brazil.

6. PLOS authors have the option to publish the peer review history of their article (what does this mean?). If published, this will include your full peer review and any attached files.

Reviewer #1: No

Reviewer #2: No

---

## [Author Response · Author response to Decision Letter 0]

31 May 2024

We are grateful to the reviewers for their time and thoughtful feedback on our work. We have carefully considered each of the points raised, addressed them in the manuscript and provide details, including line refences, below.

Reviewer #1

1. The authors performed a multivariate analysis to identify factors associated with uptake of influenza and pneumococcal vaccination. In the abstract, the authors identify significant predictors of vaccine uptake, such as residing in the Western Cape Province, but do not address the same predictors in the results section, except for in Table 3. The authors should describe in more details the significant predictors they identified.

Response: We thank you for this comment. We had not reported factors in the text part of the results section that were reported in the Table, in order to avoid repetition of the results. We have rectified this by reporting factors in the text with their odds ratios, and in the table with odds ratios and 95% CIs (lines 248-256). A description of the significant predictors / barriers is also provided in the discussion (lines 283-291). We hope this is acceptable. 

2. The authors should provide more context on why different residing in different provinces, race and education may contribute to increased odds of receiving a vaccine. The factors were highlighted in the abstract and reported in Table 3 but were not contextualized in the discussion section.

Response: Thank you for this comment. The context for socio-demographic factors identified in our study is now provided in the discussion section (lines 283-291), and we trust this is now OK.

3. The authors recommend extending free access to other vaccines, including the pneumococcal vaccine. The authors should define what “other vaccines” they believe should be free. 

Response: Thanks for this comment, which has been addressed by naming both pneumococcal and shingles vaccines (see line 486-487).

Additionally, their own data does not seem support their recommendation where they report that only 0.9% and 1.4% of individuals who did not receive the pneumococcal and shingles vaccine identified cost as a major obstacle. The authors should support their recommendations with data from their own study and/or provide more context to why the recommendation should be given considering their own results.

Response: Thanks very much for highlighting this, as we had omitted the cost obstacle results for those who were aware of these vaccines. We have now addressed this in lines 234-236. We trust this is now clear.

4. In the recommendation section the authors recommend the use of health technologies such as electronic medical records. The authors should provide the reason for this recommendation.

Response: Thank you, an explanation for this is now provided in the recommendations (lines 499-505).

5. The authors highlight a feature of the South African COVID-19 vaccination program where vaccine services were brought to the elderly. The authors should provide citations for the program.

Response: Thank you for this comment. A citation has now been included (Ref 110). 

Reviewer #2

1. Please describe the efficacy of the three types of vaccines (influenza, pneumococcal, and shingles) studied.

Response: Thank you, this information is now added to the relevant parts of the discussion section of the manuscript (lines 344-346, 382-386 and 422-425).

2. Please discuss why the rate of influenza vaccination is so low in Shanghai and so high in Brazil.

Response: This has now been added in the discussion (lines 335-339), and we trust this is now OK.

We believe that these revisions strengthen the manuscript and address the concerns raised by the reviewers. Thank you once again for the opportunity to submit our work to your Journal We appreciate your consideration of our response to the review and look forward to hearing from you soon.

---

## [Decision Letter · Decision Letter 1]

6 Nov 2024

Vaccine uptake, associated factors and reasons for vaccination status among the South African elderly; findings and next steps

PONE-D-24-06704R1

Dear Dr. Sibanda,

We’re pleased to inform you that your manuscript has been judged scientifically suitable for publication and will be formally accepted for publication once it meets all outstanding technical requirements.

Kind regards,

Vanessa Carels

Staff Editor

PLOS ONE

Additional Editor Comments (optional):

Reviewers' comments:

Reviewer's Responses to Questions

**Comments to the Author**

1. If the authors have adequately addressed your comments raised in a previous round of review and you feel that this manuscript is now acceptable for publication, you may indicate that here to bypass the “Comments to the Author” section, enter your conflict of interest statement in the “Confidential to Editor” section, and submit your "Accept" recommendation.

Reviewer #1: All comments have been addressed

2. Is the manuscript technically sound, and do the data support the conclusions?

Reviewer #1: Yes

3. Has the statistical analysis been performed appropriately and rigorously? 

Reviewer #1: Yes

4. Have the authors made all data underlying the findings in their manuscript fully available?

Reviewer #1: Yes

5. Is the manuscript presented in an intelligible fashion and written in standard English?

Reviewer #1: Yes

6. Review Comments to the Author

Reviewer #1: (No Response)

7. PLOS authors have the option to publish the peer review history of their article (what does this mean?). If published, this will include your full peer review and any attached files.

Reviewer #1: No

---

## [Editor Report · Acceptance letter]

21 Nov 2024

PONE-D-24-06704R1 

PLOS ONE

Dear Dr. Sibanda, 

I'm pleased to inform you that your manuscript has been deemed suitable for publication in PLOS ONE. Congratulations! Your manuscript is now being handed over to our production team.

Kind regards, 

on behalf of

Dr. Vanessa Carels 

Staff Editor

PLOS ONE